# Visioning a Food System for an Equitable Transition towards Sustainable Diets—A South African Perspective

**Nafiisa Sobratee** [1,*], **Rashieda Davids** [2], **Chuma B. Chinzila** [1], **Tafadzwanashe Mabhaudhi** [2,3], **Pauline Scheelbeek** [4], **Albert T. Modi** [2], **Alan D. Dangour** [5] **and Rob Slotow** [1,6]

1  School of Life Sciences, University of KwaZulu-Natal, Pietermaritzburg 3209, South Africa; chinzilac@ukzn.ac.za (C.B.C.); slotow@ukzn.ac.za (R.S.)
2  Centre for Transformative Agricultural and Food Systems, School of Agricultural, Earth and Environmental Sciences, University of KwaZulu-Natal, Pietermaritzburg 3209, South Africa; davidsr@ukzn.ac.za (R.D.); mabhaudhi@ukzn.ac.za (T.M.); modiat@ukzn.ac.za (A.T.M.)
3  International Water Management Institute (IWMI-GH), West Africa Office, Accra GA015, Ghana
4  Centre on Climate Change and Planetary Health, London School of Hygiene and Tropical Medicine, Keppel Street, London WC1E 7HT, UK; pauline.scheelbeek@lshtm.ac.uk
5  Wellcome Trust, London NW1 2BE, UK; a.dangour@wellcome.org
6  Department of Genetics, Evolution & Environment, University College, London WC1E 6BT, UK
*  Correspondence: sobrateenafiisa@gmail.com

**Abstract:** The global goal to end hunger requires the interpretation of problems and change across multiple domains to create the scope for collaboration, learning, and impactful research. We facilitated a workshop aimed at understanding how stakeholders problematize sustainable diet transition (SDT) among a previously marginalized social group. Using the systems thinking approach, three sub-systems, namely access to dietary diversity, sustainable beneficiation of natural capital, and 'food choice for well-being', highlighted the main forces governing the current context, and future interventions of the project. Moreover, when viewed as co-evolving processes within the multi-level perspective, our identified microlevel leverage points—multi-faceted literacy, youth empowerment, deliberative policymaking, and promotion of sustainable diet aspirations—can be linked and developed through existing national macro-level strategies. Thus, co-designing to problematize transformational SDT, centered on an interdisciplinary outlook and informational governance, could streamline research implementation outcomes to re-structure socio-technical sectors and reconnect people to nature-based solutions. Such legitimate aspirations could be relevant in countries bearing complex socio-political legacies and bridge the local–global goals coherently. This work provides a collaborative framework required to develop impact-driven activities needed to inform evidence-based policies on sustainable diets.

**Keywords:** agri-food system; systemic analysis; marginalized communities; sustainable diet; stakeholder engagement; interactive facilitation; multi-level perspective; deliberative policymaking

## 1. Introduction

The complexity of local and global problems challenges the agricultural, health, and socioeconomic sectors [1–3]. Moreover, the environment and biodiversity are increasingly under threat from climate change and competing development needs [4]. The food system, for instance, both threatens environmental sustainability and nurtures human health [5]. These, as well as competing societal needs, are addressed within the framework of the United Nations Sustainable Development Goals (SDGs) [6–8]. A major challenge today, and in the future, is to sustain the beneficial contributions of nature [5,9], whether from natural or managed systems [10], including food systems [11–13], to improve wellbeing for all. However, not all countries can transition towards equitable development pathways for all because of slower macroeconomic growth that reduces the pace of structural change in some countries [14].

As with its other Sub-Saharan counterparts, South Africa faces multiple biophysical, political, and socioeconomic pressures that interact to compound livelihood vulnerability, and hence limit adaptive capacity [7] Moreover, the apartheid legacy and delayed transformation suggest that new development strategies and outcomes of institutional arrangements are warranted in tackling socio-economic disparities, such as chronic poverty, household food insecurity [14,15], and other protracted socio-ecological problems [16]. Well-intended policies can lead to unintended consequences when there are incongruous policies and implementation strategies, such as skewed prioritization of economic gains over poverty alleviation, local economic development, and/or nature-based food security [17,18]. Growing evidence and, increasingly, decision-making, focus on developing the societal capacity to guide transitions that align with social and environmental alternatives. Despite having the potential to promote environmental sustainability while supporting human health and wellbeing [11,19], current trends indicate that inequalities will persist [20].

A broad range of conceptual frameworks has been applied to promote transitions towards food sustainability. Herein, we draw insights from the multi-level perspective (MLP) on socio-technical transitions [21,22], to better understand how to realize the transition towards sustainable diets amongst vulnerable, previously disadvantaged communities in South Africa. The strength of transition research is its ability to address systemic changes through long-term, multi-dimensional, and fundamental transformation processes, towards a more sustainable society [23], noting that its relevance and applicability within the agri-food sector requires an integrative approach [24,25]. The multi-dimensional concept of sustainability can cause some ambiguity as to the different normative values of food, and the tension between commodity vs. commons, which can be assisted by a more unified worldview amongst diverse stakeholders [26]. Hence, the specific objectives of this paper were to (i) examine the intricate relationships that emerged when stakeholders collectively interpreted and envisioned conceivable ways to shape a "sustainable and healthy food system" as the future desired state, against the state of the current food system; (ii) evaluate, through a scoping review, the concept of transition concerning agri-food systems; (iii) use a logical framework to demonstrate how the interventions proposed for leveraging a sustainable diet transition call for a consideration of the wider context within which the transition takes place; and (iv) identify contextual pathways to inform future policies guiding sustainable diet transitioning that take into account the influence of multiple systemic interactions and the type of actors that need to be involved.

## 2. Methods

The present work uses a mixed-method approach (Appendix A) to co-design emergent research–practice collaboration for the SHEFS program, a Wellcome Trust (UK)-funded Our Planet Our Health project, in South Africa. We applied systems thinking principles using causal loop diagramming to develop insights, make distinctions, i.e., which knowledge disciplines or institutional settings to consider, identify interrelationships and subsystems, and establish the most pertinent perspectives. Through interactive facilitation and mapping, we helped stakeholders to acknowledge and observe the complexity of interventions linked to transdisciplinary sustainability research collaboration. To unpack the complex issues linked to sustainable and healthy food systems, we aligned the interventions proposed by the workshop participants, viewed as leverage points, with the SDGs within a logical framework. Finally, we embedded these leverage points within a multiple-level perspective framework, encompassing a niche–regime–landscape continuum [27], aimed at informing the types of evidence-based policies that could potentially be devised to inform sustainable diet transitioning. The niche–regime–landscape multiple-level perspective is a prominent framework to analyze socio-technical transitions towards sustainability, which stems from evolutionary economics and the social construction of technology [27]. Central to this is that economic processes evolve and that economic behavior is determined both by individuals and society as a whole [28]. In the present context, sustainable socio-technical transition,

therefore, refers to new kinds of agri-food systems shifts, and the types of actors required to support participatory consensus outcomes that encourage desired change.

Thus, the paper uses an exploratory approach to examine the co-design process applied at the beginning of the SHEFS project. The emergent issues raised during the interactive facilitation, consisting of the stakeholder meeting and de-briefing analysis, were used to guide further interdisciplinary evidence building throughout the project. To enhance the robustness of this paper's outcome, insights [29] from scientific realism are used [30]. Thus, on the one hand, the bibliometric method is applied to corroborate with the systemic relationships identified in the causal loop analysis, i.e., from the stakeholder engagement process. On the other hand, the systematic review is used to support the argument for creating a sustainable diet innovation system (SDIS) based on the premise that sustainable diet transitioning (SDT) can be viewed from the socio-technical perspective. The papers reviewed in the scoping review aid in building the framework in the current paper. Viewed together, the causal loop analysis, bibliometric analysis, scoping review, and multiple-level perspective of the transition systems theory serve as a means of triangulation to conceptualize the emerging and co-evolving issues that need to be considered to inform policies on SDT.

Ethical approval was granted by UKZN, and all participants provided informed consent for their participation.

### 2.1. Systemic Analysis of Sustainable Diet Drivers

We captured the outcomes of the first Sustainable and Healthy Food Systems (SHEFS) Program key stakeholder workshop to define the current and desired state of the agriculture, environment, and social system in South Africa. The facilitated workshop brought together stakeholders ($n = 39$) from key government competencies, across the three levels of government policymakers and practitioners (municipality, provincial, and national), as well as academics and post-graduate students from crop science, food security, nutrition, health sciences, development studies, environmental science, and biodiversity conservation. For the systems-approach-based interactive facilitation exercise, the targeted sample size, $n$, was 50. However, some of the participants could not attend the workshop. The shortcomings, if any, were counteracted during the peer debriefing session on the following day.

To facilitate the process, participants were asked to consider, firstly, SHEFS's (https://shefsglobal.lshtm.ac.uk/ accessed on 3 March 2022) overarching aim: "to provide policymakers with novel, interdisciplinary evidence to define future food systems policies that deliver nutritious and healthy foods in an environmentally sustainable and socially equitable manner" as a guiding star, which is a preferred future state of the system. Secondly, a "near star" question was asked: "What is the effectiveness of the current food-crop-environment-health system for addressing human livelihoods and welfare, considering knowledge, understanding, legislation, policies, implementation, and sustainability?" For this exercise, the workshop participants spent 3 h in groups, each including representatives of all stakeholder types, to brainstorm and map (i) the state of knowledge, and (ii) the possible desirable states.

We then wanted to capture a systemic overview from each group, through causal diagrams, about how the stakeholders' mental models related to the SHEFS program's overall objectives. Following a briefing on the conventions of drawing interrelationship digraphs (concept terms connected by a bi-directional line) [31] and causal loop diagrams [32], the participants in each group were then asked to respond to the questions by drawing their group's collective interpretation of the system (without idea exchange amongst groups). All diagrams generated were refined by engaging with the participants through interactive facilitation during the workshop to ensure that the ideas were accurately captured and representative, and, thereafter, updated by the author team to produce conventional causal loop diagrams (CLDs). CLDs are used to conceptually model dynamic systems, which can be social and/or ecological, by mapping how variables, i.e., factors, issues, and processes, influence one another, [33]. Common variables that appeared in the different group

diagrams were identified, and the nature of their causal relationships was highlighted to create interlinkages among the sub-systems, uncover any underlying feedback structures, and identify leverage intervention points in the system [34,35].

The following day, we conducted post-workshop expert deliberations, including the principal SHEFS investigator and nutrition expert (AD), the principal investigators in the environment (RS) and crop (AM) fields, the project coordinator for South Africa (RS), one researcher in diet and health (PS), and two researchers representing the health sciences co-investigator. Collectively, we acted as key informants to identify science–action interventions, from the previous day's outcomes, with high leverage impacts for biodiversity (Nature) and end-user beneficiaries (People). During a five-hour focus group discussion, we interrogated the linkages and nature of the different sub-systems identified the previous day to develop a strategic framing. The causal loop diagrams were reviewed by the experts with the workshop facilitator (NS) and complemented by (i) groundwork that was already being undertaken by the researchers, as well as (ii) additional potential research gaps capable of delivering sustainable diet leverages that had not been identified the previous day but emerged from the interrogation of the linkages and causal loops. Causal loop analysis was performed and, where relevant, system archetypes [36] were applied to present a systems view of the interplay between the different forces identified. Color coding based on subsystems, identified archetypes, and/or inter-linkages was then used to enhance the representation of the diagrams. Relevant literature was used to substantiate, align, and unpack the interpretations of the stakeholder views concerning the guiding star and near star questions.

### 2.2. Review of Bibliometric Studies on the Sustainable Transition of Food Systems

2.2.1. Review of Multi-Level Perspective in Food Agri-Systems

The emergence of persistent environmental degradation worldwide has raised the question of how to induce a societal transformation towards more sustainable production, consumption, and biodiversity protection [37]. New technologies or governance approaches, economic deregulation, and changes in consumer behavior have been introduced to relieve urgent problems [38,39]. However, generally, transformational processes are slow or even failing, technology diffusion is inefficient, governance concepts are implemented in theory only, deregulation causes high uncertainties, and consumers do not act as anticipated [37]. A broad range of frameworks has been used to explore the transition towards sustainability [24], such as the multi-level perspective (MLP) on socio-technical transitions [40], transition management [41] (TM), strategic niche management [42] (SNM), technological innovation system [43] (TIS), and the social practice approach [25] (SPA). MLP argues that transitions, i.e., large-scale socio-technical change, occur through interactions between processes at three levels. First, niche innovations build up impetus through knowledge production processes, such as research and/or performance improvements, and support from powerful civil society groups. Herein, the concept of 'experimentation' occupies a central position within the academic component that investigates transformations towards sustainable socio-technical systems. This focus on experimentation is a key agent of change that sets the sustainability transitions field apart from the wider literature of social change and policy theory [23,44]. 'Socio-technical experimentation' can be contrasted with the notion of experimentation used in the natural sciences. It implies a more engaged and social constructivist position, whereby society is itself a laboratory and a variety of real-world actors commit to the messy experimental processes tied up with the introduction of alternative technologies and practices, to purposively re-shape social and material realities [44,45]. Second, the concept of the socio-technical regime has been formulated to account for the delay and path-dependency experienced in articulating and understanding transformative change [46]. Regimes, therefore, result from the co-evolution of institutions and technologies over time, which become positioned in practices and routines. Sociologists of technology refer to regimes as consisting of a variety of actors, that is, scientists, policymakers, consumers, and special-interest groups that contribute to the

patterning of technological development [47]. The sociotechnical regime concept, therefore, accommodates a broad community of social groups, and their alignment of activities and their interactions result in the stabilization of socio-technical trajectories in many ways: Regulations and standards, the adaptation of lifestyles to technical systems, investments in machines, infrastructures, and competencies [48–50]. Third, the socio-technical landscape, which could be macro-economics, deep cultural patterns, or macro-political developments, constitutes an exogenous environment beyond the direct influence of niche and regime features [24]. Changes at the landscape level usually take place over decades, and such changes can exert pressure on the regime through a selective process of societal change—sectoral policies, education system, and market-driven technological novelty—and create windows of opportunity for regime change, subsequently providing leverage for niche innovations to emerge and create a new regime [51]. A transition therefore occurs when a regime is transformed as it responds to systemic changes. The MLP framework is useful in understanding contexts that have co-evolutionary properties as it aids in justifying the importance of an adaptive policy approach when addressing complex problems burdened with intrinsic dynamics [50].

We conducted a search in the Web of Science Core Collection (SCI-EXPANDED, SSCI, A&HC1, ESCI), dated 3 August 2021, for publications on sustainable food transitions, more specifically those that applied the multi-level perspective. The output was narrowed down to include articles that deal with the "food systems" topic. Hence, the search term used was:

TOPIC: (socio-technical transition AND multi-level perspective) AND TOPIC: (food systems).

The search identified 15 articles (*n* = 15), and given the small sample size, all were retained for scrutiny. In the analysis of the output, the following attributes were derived: The context of the transition research, the transition process, any specific methodologies/approaches, and the action domain that emerged.

### 2.2.2. Developing Multi-Level Insights for Sustainable Diet Transition from the Stakeholder Systemic Analysis

Having explored MLP transitions in the literature, we then used the output from the stakeholder workshop within the MLP to showcase how evidence-based sustainable diet policies can be rendered more effective in addressing barriers and opportunities, thereby realizing sustainability transition in the near future. We identified examples of interventions that were co-designed by stakeholders and assessed by the expert deliberations as leverage points within the socio-technical and socio-ecological context. We then categorized those as proposed policy measures against the niche–regime–landscape (micro-level, meso-level, meta-level, respectively) continuum and described the network of actors responsible.

## 3. Results

### 3.1. Impact of Extrinsic Systemic Issues on Small-Holder Farming (SHF) in South Africa-A Nation-State Level Perspective

Participants were questioned whether the end of apartheid had improved the situation for the South African smallholder farming sector, which is essentially comprised of the previously disadvantaged population. It was agreed that this sector remains seriously limited and poorly structured, being embedded in a reinforcing vicious cycle (R1) (Figure 1) that undermines capacity-building for sustained and diverse local food production. Not only is the smallholder farming sector disadvantaged from a productivity standpoint, but the institutional dynamics related to the socio-economic conditions render it nearly impossible for emerging farmers to thrive [52,53]. Loop B1 (Figure 1) describes how the economic transformation policy agenda aims to reduce the current limitations of the historically underprivileged smallholder food producers through an emphasis on sectoral development planning as is elaborated in the National Development Plan [54]. Under apartheid rule, the relative economic outcome beneficiated the privileged societal group (Loop R2) to the detriment of the historically underprivileged group (Loop R3) (Figure 1). With the advent of democracy in 1994, the objective of the transformative agenda was

to redress privileged beneficiation by opening the economy and progressively offsetting the unequal economic outcomes. In the new South African Constitution, 'every citizen is equally protected by law' and all are obligated to 'heal the divisions of the past' while 'recognizing injustices of the past' [55]. Yet, inequality is still pervasive [56]. More than two decades after the dispensation, and despite the various initiatives of the economic transformation processes, the governance and reality of the smallholder farming sector at large still grapples with systemic limitations, as represented in the balancing loop B2 (Figure 1).

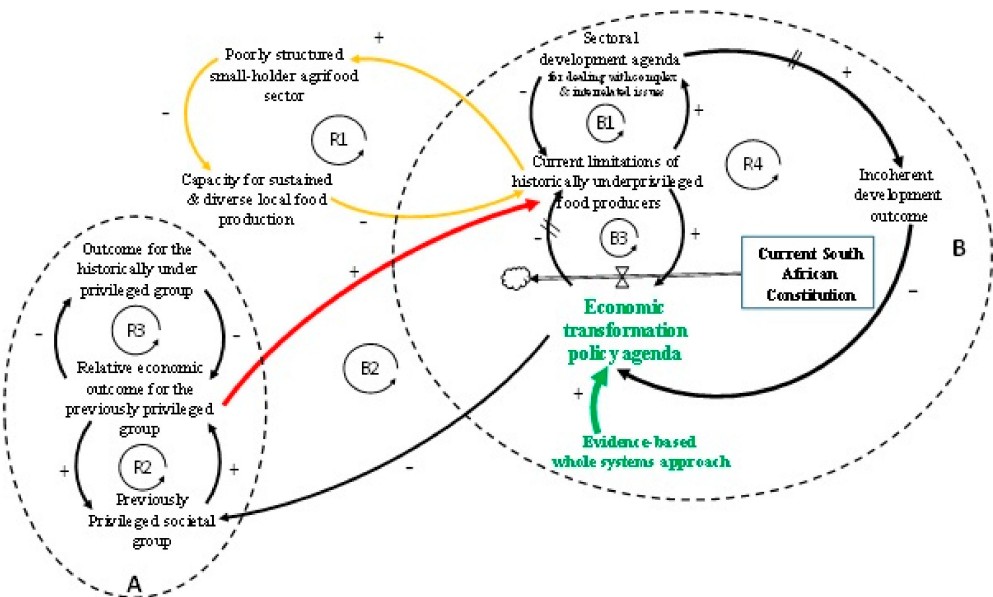

**Figure 1.** Interlinkages and causation pathways impacting current limitations faced by historically underprivileged food producers. (**A**) A 'Success to the successful' archetype that explains South Africa's previous segregated socio-economic situation. (**B**): A 'Shifting the burden archetype', whereby an over-emphasis on sectoral development with the insufficient implementation of whole-systems evidence-based approaches results in incoherent development outcomes that eventually undermine the achievement of the economic transformation. Red arrow: An anomaly that persists for more than two decades among the previously marginalized despite the end of apartheid rule. Ideally, the current limitations of the historically underprivileged should have decreased, given the inclusive and/or affirmative opportunities offered through the new Constitution. This means that development solutions that are being brought about are not tackling the root causes of problems, resulting in inadequate outcomes, e.g., the small-holder agri-food sector remains poorly structured with insufficient capacity for a thriving local production (Loop R1 with yellow arrows: The vicious reinforcing consequence that perpetuate limitation in the smallholder food sector). Arrows with a double dash: Intrinsic systemic delays characteristics of complex systems; hourglass symbol: Dynamic nature of the variable represented as a rate of change, herein indicating that the economic transformation ought to be an on-going adaptive process, driven by the democratic Constitution that acts as a guiding principle for a new normal in South African politics to influence governance and steer other multi-dimensional change.

### 3.2. Efficacy of Structural Adjustment for Socio-Economic Upliftment
#### 3.2.1. Transitioning of Smallholder Farmers

As part of unpacking loop B2 (Figure 1), and especially the associated causal factors, participants referred to the unintended consequences of agricultural policy. This was based on the premise that creating opportunities for the previously disadvantaged to own farmland would help subsistence farmers' attempts at commercialization, and create a middle group termed the 'emerging farmer' sector. Policies enabling the shift from smallholder farming (SHF) to small-scale commercial farming (SCF) have had two types of spill-over effects

(Figure 2). The first one creates a causal pathway with desirable effects whereby the farmers who have enough leverage to invest can improve their socioeconomic status. This would alleviate their poverty level by improving their flow of household revenue (Loop R5). Subsequently, they can improve their living standard, and, in effect, ensure access to fresh and convenient foods. The variable 'consumer's revenue' here refers to the previously disadvantaged population that are also food consumers in the SHF system.

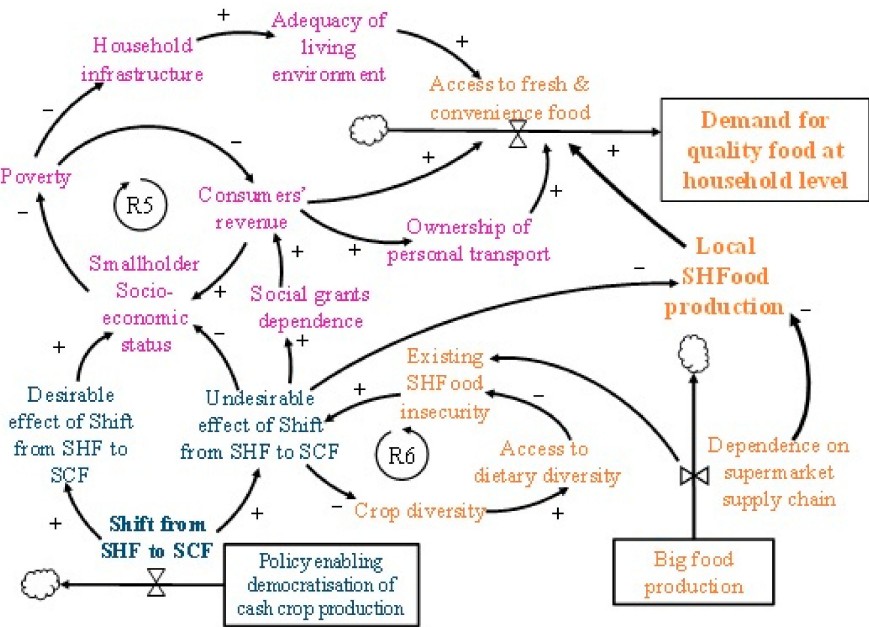

**Figure 2.** Impact of the shift from small-holder farming (SHF) to small-scale commercial farming (SCF) on small-holder socio-economic status. Reinforcing loop R5: The spill-over effects resulted in desirable and undesirable effects on the socio-economic status of the smallholder farmers, which either relieve or exacerbate poverty level depending on how successful they emerge as small-scale commercial farmers. Reinforcing loop R6: Unsuccessful cash crop ventures diminish access to dietary diversity and worsen food insecurity, such that eating habits are linked to key issues around affordability and convenience. Orange variables: Smallholder/previously under-privileged access to food and dietary diversity. Blue variables: Effect of policy enabling the democratization of cash crop production. Purple variables: Socio-economic realities of the smallholder sector.

The second effect is when the farmers, despite aspiring to farm successfully, still find their socio-economic and household food security status undermined. This occurs due to a combination of factors [57,58] such as a poor business framework and insufficient input support, know-how, and infrastructure, hence contributing to an undesirable effect on the shift from SHF to SCF. The example of cash crop production, such as sugarcane in the KwaZulu Natal Province, was used to illustrate the unintended consequence in the reinforcing vicious loop, R6. In striving to produce sugarcane as a monocrop, on-farm crop diversity is reduced because food crops are neglected. Dietary diversity within such households, which depends on subsistence farming, is undermined, leading to household food and nutrition insecurity. These consumers must increasingly rely on the 'Big Food Industry'. Such a type of food sourcing from supermarket outlets creates a dependence on supermarket supply chains, which is unaffordable and inaccessible to poor communities, further exacerbating existing household food insecurity.

3.2.2. Impact of Socio-Economic Conditions on Access to a Healthy and Sustainable Diet

When the socio-economic status of smallholder and underprivileged communities result in sub-optimal revenue, poverty remains rampant and pervasive. The ubiquitous prevalence of poverty creates dependence on social grants to support household revenue for consumption. This dependence is counter to other policy decisions, such as improving

smallholder socio-economic status through economically sustainable means. The 'adequacy of the living environment', itself dependent on revenue generation, is a critical factor that prescribes the type of food consumed (Figure 2). The poor and previously underprivileged communities occur in the peri-urban region as sub-organized settlements or as informal segments in the metropolitan cities. It is only when adequate revenue is allocated towards household infrastructure and facilities, such as access to electricity and the ability to store perishable and/or convenience food in a refrigerator, and the ownership of car or access to another form of transport, that access to food can be definite at the household level.

Moreover, participants referred to the fact that the ways the previously under-privileged people consume food culturally, and the historically conditioned meanings ascribed to food and eating, must be considered to understand how to shift current food consumption towards a sustainable transition. The emerging patterns [59] consist of a preference for cheap grain staples, sugar, soft drinks, and chicken, frequently sourced through informal channels. This implies that, apart from price and convenience, the symbolic and aspirational domain of food aesthetics and the social functions of visible consumption become key forces shaping food choices. Currently, individual preferences and attitudes are stronger determinants of food choice, rather than sustainable food choice for well-being acting as determinants of food choice (Figure 3). This is a consequence of the increasing individualization of society, an outcome of western lifestyle fast-food aspirations. When it comes to food choice and consumption, on the one hand, the individualization of lifestyle and lavish food preferences represent the fulfillment of historically unfulfilled desires dating from apartheid rule. The resulting attitude renders 'past foods', mainly maize porridge and vegetables, as an undesirable reminiscence of the 'difficult past', and healthy food is perceived as unappealing or too expensive. On the other hand, the sprawl of informal settlements and abject poverty leads to poor food choices due to financial constraints and the inability to afford healthy food [Poverty → Food choice for well-being (Figure 3)]. Both situations are not aligned with food choices that promote well-being.

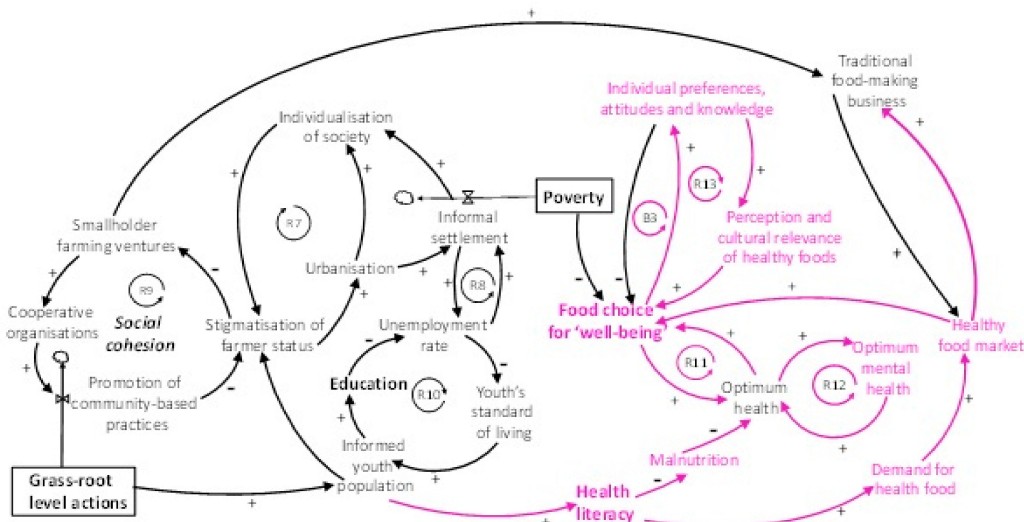

**Figure 3.** Socioeconomic factors that impact small-holder farming ventures and 'Food choice for wellbeing'. Blue variables: Spill-over effect of de-agrarization through stigmatization leading to the proliferation of informal settlement and unemployment. Pink variables: Influence of 'health literacy' in leveraging food aspirations and 'food choice for well-being'. Stimulating smallholder farming ventures and driving the demand for a healthy food market ought to stimulate traditional food-making businesses, which could then influence positive feedback upon food aspirations and choice. 'Traditional food making business' emerged as a currency to stimulate both smallholder farming ventures and to create a drive for the healthy food market and eventually 'Food choice for well-being'.

The individualization of lifestyle is the outcome of a spill-over effect resulting from a vicious reinforcing loop involving stigmatization of farmer status and urbanization, as

seen in R7 (Figure 3). Participants discussed how, despite the political will for a more inclusive agricultural economy, smallholder farming has been on the decline in recent years because of a combination of macroeconomic constraints. In particular, the stigmatization of farming activities has discouraged youth participation in agriculture [60,60]. Post-apartheid de-incentivization of agriculture was deemed as the major systemic barrier that deterred communities from sustaining small-holder farming. Coming from a difficult past characterized by restrictions on movement, education, wealth accumulation, among other things, the palpable post-apartheid response has seen an increase in movement, leading to a rural exodus because of the perceived opportunities and prosperity that the urban regions could potentially provide.

*3.3. Interventions to Leverage Sustainable Diet Transition*

3.3.1. Socio-Economic Factors, Social Aspirations, and Individual Food Choice Behavior

Participants posited that high-leverage interventions would necessarily have to include improving the health literacy of consumers to tackle problems of malnutrition, and to create the demand for healthy food. Therefore, instead of "Individual preferences, attitudes and knowledge" influencing whether consumers opt for "Food choice for well-being", participants proposed that the sustainable diet transition should be stimulated in such a way that "Food choice for well-being" becomes the determinant for food preferences and attitudes. Participants emphasized that strategies to shift from meat-centered dishes to a variety of healthy dishes might not be viewed by consumers as authentic and convenient food. This is because meat alternatives might not be viewed as aligned with the post-apartheid freedom of choice lifestyle, which is intrinsically linked to self-determination realities [61]. As such, individual choice is a complex dietary behavior and is influenced by various physiological, social, and cultural factors [62,63]. Therefore, taste profiles should be taken into account when proposing healthy and sustainable menus and meals [63,64]. In Figure 3, this is represented as the balancing loop, B3. As a result of the "Food choice for well-being" → "Individual preferences, attitudes and knowledge" relationship, a desirable and aspired-to loop is created as R13.

Table 1 explains the causation pathway from the proposed interventions to the expected outcomes. The UN SDGs are used to provide the overarching context and relevance of the transformative trajectory.

**Table 1.** Transformative pathways to influence food-related social aspirations towards sustainable and healthy food pathways.

| Interventions | Causal Pathway | Expected Outcomes | Relevance as Functionally Interrelated SDG Targets | |
|---|---|---|---|---|
| Socio-economic factors, social aspirations, and individual food choice behavior | | | | |
| Mobilize cross-sectoral resources to promote sustainable diet choices through health literacy | Health literacy → R11: Pathway to influence food choice that promotes health and well-being | Health literacy to reduce malnutrition and improve health, including mental health | | *T2.2* End all forms of malnutrition |
| | | | | *T3.4* Reduce mortality from non-communicable diseases and promote mental health |
| | | | | *T4.6* Universal literacy and numeracy |
| | R12: A reinforcing loop that highlights the holistic nature of health as comprising of both physiological health and mental health | Diet and lifestyle based on "Food choice for well-being" | | *T12.8* Promote universal understanding of sustainable lifestyle |
| | | | | *T8.3* Promote policies to support job creation and growing enterprises |

**Table 1.** *Cont.*

| Interventions | Causal Pathway | Expected Outcomes | Relevance as Functionally Interrelated SDG Targets | |
|---|---|---|---|---|
| Support growing traditional and healthy food-making | Spill-over effects of boosting small-scale farm ventures to promote healthy traditional food-making | Driving consumer demand to create a market for healthy local food and support agri-food entrepreneurship | | *T1.1* Eradicate extreme poverty food |
| Foster pro-poor food choices for high-quality sustainable diets | R13: A desirable and aspired reinforcing loop which only occurs if 'food choice for well-being' can influence 'Individual preferences & attitudes' | 'Food choice for well-being' habit positively impact 'Individual preferences and attitudes', which can then lever 'Perception & cultural relevance of healthy foods' | | *T 2.1* Universal access to safe and nutritious |
| | | | | *T 10.2* Promote universal socio-economic and political inclusion |
| | | | | *T12.8* Promote universal understanding of sustainable lifestyle |
| | B3: An important goal-seeking loop to improve preferences & attitudes which cannot be achieved without the 'health literacy' causal pathway and outcome of loop R11, to then, link 'Food choice for well-being→ Individual preferences & attitudes'. B3 is however compounded by poverty level. | Successful behavior change provided food choice determinants such as poverty level and therefore access to food, are tackled. A pro-poor sustainable lifestyle would counteract individual preferences and attitudes which do not align with healthy diet pathways | | |

1. Loops and variables unpacked are from the causal loop diagram in Figure 3. Causal pathways are relationships that are anticipated to generate expected outcomes; impacts of interventions could occur through different pathways but eventually share the same overarching sets of UNSDG outcomes. The relevant United Sustainable Development Goals (UNSDGs) targets are from GOAL 2: Zero Hunger; GOAL 3: Good health and well-being; GOAL 4: Quality Education; GOAL 10: Reduced Inequality; GOAL 12: Responsible Consumption and Production.

### 3.3.2. Reinforcing the Democratization of Knowledge to Unleash Sustainable Diet Transitions

Based on the types of interventions endorsed by the participants, the theme of education emerged as a common enabling concept in addressing the limitations of the smallholder sector concerning sustainable diet transitions and environmental conservation. Functional education could leverage the implementation of sustainable income-generating community-based interventions to promote food security and sustainable beneficiation of natural capital from agriculture and related novel entrepreneurial activities. Participants referred to the Strategic Plan for South African Agriculture [59], dated as far back as 2001, which has already aimed to increase the incomes of the poorest groups in society through opportunities for small-/medium-scale farmers. In effect, the National Department of Agriculture [65] pays particular attention to small-scale agriculture with three strategic aims: (i) Making the sector more efficient and internationally competitive, (ii) supporting production and stimulating an increase in the number of new small-scale and medium-scale farmers, and (iii) conserving agricultural natural resources. However, these aims are yet to gain adequate leverage, [66] and, hence, are still relevant as expected outcomes of multi-lateral evidence-based interventions in achieving sustainable and healthy food systems. Environmental literacy and agri-food literacy were deemed as important drivers to leverage new types of ecosystem services through inclusive social innovation (Figure 4). The example of reduced crop diversity as an outcome of sector-based thinking in policy planning was mentioned again by participants. In this instance, the lens of coherence in land-use planning was used to explain how a change in land-use patterns (Linkage between 'Change in land use

pattern for monocropping' → 'Crop diversity', Figure 4), caused by avocado, sugarcane, and agroforestry, when unchecked, can jeopardize crop and food plate diversity. Therefore, ongoing evidence synthesis on environmental change (Loop B5) in local sustainability experiments would be important in understanding how to alleviate cross-cutting issues and unintended consequences arising from sectoral policy decisions.

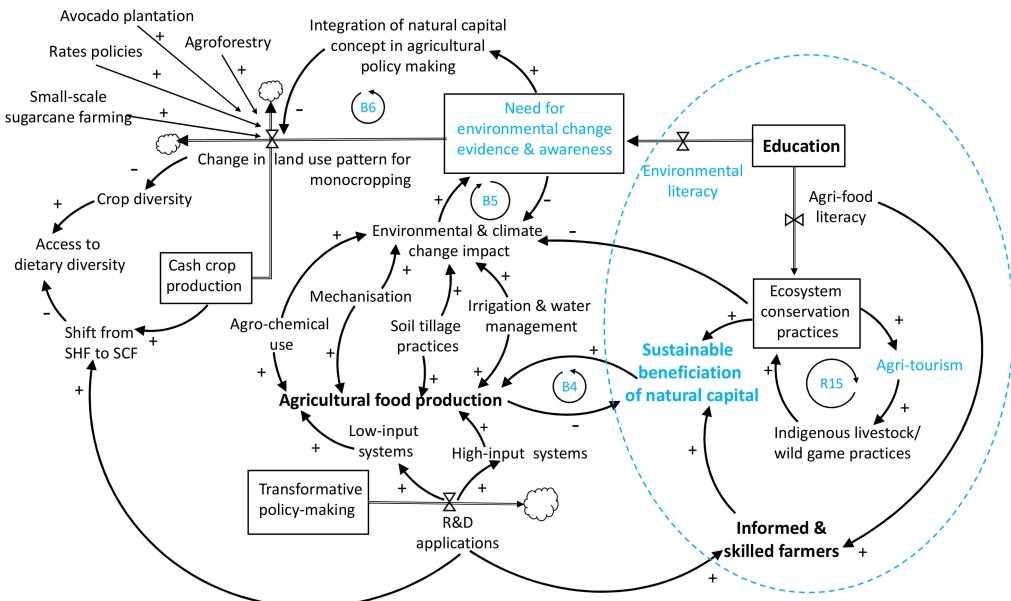

**Figure 4.** The emergence of functional education as a key nexus to improve environmental awareness and promote agri-food literacy. Agricultural food production in South Africa is essentially a dual system consisting of the low-input subsistence system and the high-input commercial systems that are stimulated through ongoing R&D policies. Agricultural operations (mechanization, irrigation and water management, soil tillage practices) and input (agrichemical use) applied to boost food production worsen climate change and environmental impacts and have a dampening effect on the sustainable beneficiation of natural capital (Balancing Loop B4). Emphasis on multi-faceted literacy ought to incentivize social entrepreneurial innovation and sustainable beneficiation of natural capital from agricultural activities (Loop R15: Virtuous reinforcing loop where the variables mutually reinforce agritourism, indigenous practices, and promote ecosystem conservation practices; all stimulated through functional agri-literacy). Loop B5: Evidence-building and awareness can reduce the impact of agricultural activities. Loop B6: On-going evidence-building and creation of awareness regarding environmental change ought to influence multi-sectoral policymaking, for instance by framing natural capital as transformational.

Ideally, the democratization of knowledge ought to strengthen bottom-up actions, e.g., in the form of cooperative organizations and civic actions, to deliver greater awareness of policy incentives to community members (Figure 5). Moreover, the inclusion of curriculum and governance components that enable the formalization of the Indigenous Knowledge System (IKS) ought to complement mainstream education, to enhance the ongoing development of the much-aspired knowledge-based economy (Loop R16). Participants emphasized that to provide a consolidated frame of action to such an endeavor would require the inclusion of a vibrant policy process that is designed to be adaptive in accommodating IKS (Loop 17). An improved organization of democracy and civic interest could create sufficient grounds to render the education system more contextually functional, improve employment relevance for youth, and, consequently, their standard of living. The ability to make an informed choice would further motivate the pursuit of appropriate information and enhance the subjective appropriation of their life course based on sustainable well-being tenets [67], amongst others. Such a course of action would enable youth to transition into responsible citizenship. Young individuals will have garnered a better understanding

of individual responsibility concerning the different dimensions of sustainable well-being, for instance in terms of diet and health choices [68], and the shaping of environmental civic engagement within communities [69]. Table 2 displays the transformative pathways capable of leveraging sustainable and equitable food security from a knowledge economy perspective. It shows, amongst others, the comparative advantage of including IKS in policies. This could become an opportunity to adjust the general concept of the innovation system to local contexts and practices and include bottom-up socio-ecological approaches to create a stimulus for biodiversity and conservation-friendly entrepreneurial and social innovation. The expected outcomes would have direct relevance to several UN SDGs as shown in Table 2.

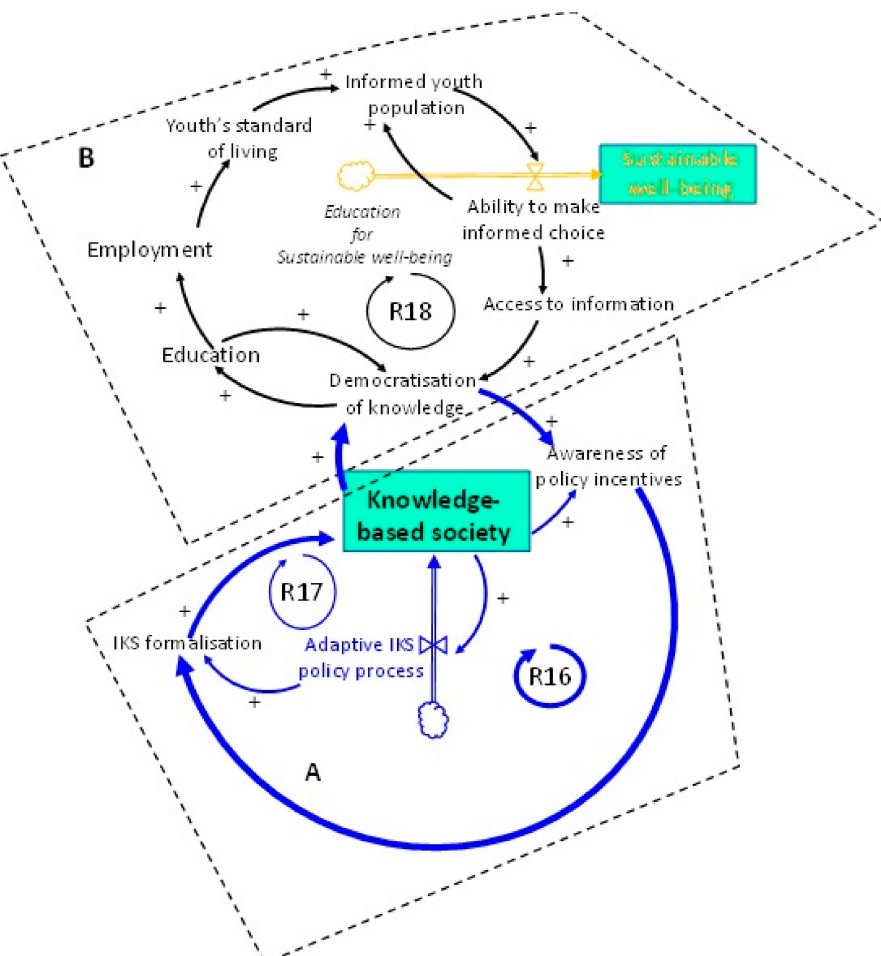

**Figure 5.** Interventions capable of driving sustainable well-being from the education perspective. (**A**) Reinforcing loop, R16, on the consolidation of knowledge-based society by inclusion of Indigenous Knowledge System (IKS) formalization, mediated by the democratization of knowledge and awareness of policy incentives. Loop R17: An adaptive IKS policy process reinforces the inclusion of indigenous cultural capital and knowledge; (**B**) tackling unintended effects of social exclusion of youth by using education as a mechanism to enable and drive responsible individual choice for sustainable well-being.

**Table 2.** Transformative pathways to leverage sustainable and equitable food security from a knowledge economy perspective. [2].

| Interventions | Causal Pathway | Expected Outcomes | Relevance as Functionally Interrelated SDG Targets | |
|---|---|---|---|---|
| Implementation of sustainable income-generating community-based interventions to promote food security and alleviate poverty for the marginalized within a knowledge economy perspective | Environmental & Agri-food Literacy with Loop R15: Reinforcing virtuous loop where agri-food and environmental literacy could leverage the development of joint entrepreneurial ventures to boost indigenous livestock & wild game practices | Use capabilities of functional education to create a stimulus for biodiversity and conservation-friendly entrepreneurial and social innovation Uplift, promote, and preserve indigenous conservation practices and know-howSustainable use of ecosystem services as innovation instruments to reduce social inequality | | *T4.6* Universal literacy and numeracy |
| | | | | *T8.6* Promote youth employment, education, and training |
| | | | | *T8.9* Promote beneficial and sustainable tourism |
| | | | | *T9.3* Increase access to financial services and markets |
| | | | | *T13.3* Build knowledge and capacity to meet climate change |
| | | | | *T15.* An Increase financial resources to conserve and sustainably use ecosystems and biodiversity |
| Education as vehicles for sustainable development actions | R18: 'Education for sustainable well-being' to elevate the youth's standard of living and knowledge base | An educated youth would cultivate the capacity of discernment for: satisfaction with sustainable lifestyle and built environmentemployability & entrepreneurial opportunities awareness adequate access to information to be informed on food choice and guide individual aspirations | | *T4.7* Education for sustainable development and global citizenship |
| | | | | *T8.3* Promote policies to support job creation and growing enterprises |
| | | | | *T8.6* Promote youth employment, education, and training |
| | | | | *T9.C* Universal access to information and communications technology |
| | | | | *T10.3* Ensure equal opportunities and end discrimination |
| | | | | *T12.8* Promote universal understanding of sustainable lifestyle |

**Table 2.** *Cont.*

| Interventions | Causal Pathway | Expected Outcomes | Relevance as Functionally Interrelated SDG Targets | |
|---|---|---|---|---|
| **Promotion of informational governance** | R9: Reinforcing virtuous loop aimed at strengthening collective actions through cooperative and social organizations to promote LED | Strengthening of social cohesion through grass-root level actionsCapacitate dignity and identity construction to advocate a novel idea around the status of rural and/or peri-urban farming | | *T8.2* Diversify, innovate and upgrade for economic productivity |
| | | | | *T10.2* Promote universal social, economic, and political inclusion |
| | | | | *T16.7* Ensure responsive, inclusive, participatory, and representative decision-making at all levels |
| | R16: IKS based policies to improve implementation coherence in a knowledge-based economyR17: Raising awareness of the advantages of policy incentives ought to boost the formalization of IKS through an adaptive process | Creation of IKS-based comparative advantages and contextual rationale for positive societal change in the previously marginalized communities | | *T10.2* Promote universal socio-economic and political inclusion |
| | | | | *T11.3* Protect the world's cultural and natural heritage |
| | | | | *T16.6* Develop effective, accountable, and transparent institutions at all levels |
| | | | | *T16.7* Ensure responsive, inclusive, participatory, and representative decision-making at all levels |

[2] Loops and variables unpacked are from the causal loop diagrams in Figures 3–5. The interventions and their impact indicated through causal relationship(s) are described. The outcomes created for a successful transition towards sustainable diet transition are shown with the relevant United Sustainable Development Goals Targets (https://www.globalgoals.org/resources accessed 3 March 2022). Main goals are-GOAL 1: No Poverty; GOAL 2: Zero Hunger; GOAL 4: Quality Education; GOAL 8: Decent Work and Economic Growth; GOAL 9: Industry, Innovation, and Infrastructure; GOAL 10: Reduced Inequality; GOAL 13: Climate Action; Goal 15: Life on Land; GOAL 16: Peace and Justice Strong Institutions. Causal pathways are thought to generate expected outcomes; impacts of interventions have different pathways but can have the same overarching sets of outcomes as per the SDG Targets; LED: Local Economic Development.

### 3.4. Mobilising Systems and Coalition of Actors for Sustainable Diet Transition

### 3.4.1. Review of Food Systems Sustainability Transition

The use of transition systems research in agri-food systems [70,71] becomes prominent when the problem is complex, ambiguous, and requires the concerted action of many different types of actors to make transformation processes effective. The dialectic relationship between stability (i.e., established rules, governance, habits) and desired and feasible change in understanding how the transition occurs is central. There are multiple interpretations of what is to be sustained and what is to be developed when considering any socio-technical system. This is because there are multiple goals and pathways for development, but, in practice, only a subset will be fully pursued. Knowledge is also socially constructed, and politics of power influence explain why some systems or certain sustainability goals tend to be prioritized. In the MLP framing, the concept of "local sustainability experiments" is used to describe what would be the sectors and actors co-existing and operating at the niche level to create novelty. When the unit of analysis lies in sociotechnical systems, the analysis involves a wide range of actors, and no agent has full accountability or ownership of sociotechnical systems. The novelties can be a combination of scientific research or civil society actions that generate evidence for change.

In agri-food systems, the multiple-level perspective is useful to empower communities to generate grass-root and social innovations [72]. As such, it is a long-term process, spanning decades, characterized by uncertainty and open-endedness. In effect, sustainability

journeys are intrinsically dynamic as there are multiple transition pathways, which implies multiple values, and disagreement, since the sustainability notion is highly contested [73]. To catalyze desirable changes in such a context, public policy [74,75] plays a central role in shaping the sustainability transition. As a means to support evidence-based understanding of transition transformation whereby the different dimensions of socio-technical systems transitions are considered, various research constructs are used as methods and/or approaches [76] such as systems thinking [77], system diagnosis [78], retroduction [76], scenario analysis [79], and critical realism, and are applied to design the interdisciplinary space that requires action.

3.4.2. Empowering Vulnerable Communities to Achieve Sustainable Diet Pathways

Table 3 illustrates how the leverage points can be developed to generate evidence capable of stimulating the policymaking process. Based on the multi-level perspective of the socio-technical transitions, the proposed leverages are expressed as policy measures that could be developed, and the categories of actors that could influence the cross-scale transformation process identified. Thus, using the reference of the overarching objective of the SHEFS program, society is viewed as a set of overlapping socio-technical systems consisting of networks of actors such as consumers, environmental action partnerships, small-scale food producers/farmers, socio-cultural/non-governmental organizations, value chain financing specialist, and youth/women groups, who act upon institutions, cultural practices, and knowledge. Much emphasis is placed on developing substantiative equality, given the socio-political legacy of South Africa. At niche levels, this can be achieved through local experiments on agri-food systems, not only as a science but to unleash capabilities, empowerment, inclusivity, and embrace the socio-ecological viewpoint. For instance, at the time of conducting the current workshop, the Neglected and Underutilized Species (NUS) component of the project had started to generate evidence through scoping reviews and multi-criteria suitability analysis, which subsequently informed a policy brief [80–83].

Because agents/stakeholders with different behavioral characteristics play a role in the distinct stages of transitions, notably pre-development, take-off, acceleration, and stabilization (establishing the change over time) [84], they influence the transition process through their goals, knowledge, information, power, interactions, relations, and interests. Thus, for instance, regime-level policy measures that need to be designed to advance rural agritourism as a development tool must consider new transformational challenges. For agri-tourism to exist, it not only requires mastering ecosystem conservation and indigenous wildlife practices, but entails a seamless harmonization with rural entrepreneurship processes to become transformational transitions [85,86]. Criterion 8 of the IUCN standard emphasizes the need to learn from the implementation of nature-based solutions (NbS) to 'trigger transformative change' [87]. However, for this to be realized, NbS must be framed as transformational. The framing of an issue is a key point of focus in transformations, as it influences how people understand the topic itself, shaping how problems and solutions are defined and addressed [88,89]. To catalyze change, the drive for successful transition can be addressed by beginning with developing policies with positive reinforcing loops between the niche (micro-level triggers) and the window of opportunities provided at the landscape (macro) levels.

**Table 3.** Multi-dimensional and multi-scalar interactions among the sustainable diet transition sectors, technology, markets, policy, and culture, capturing the complexity of systematic changes towards sustainability.

| Level | Policy Measure | Example of Interventions That Can Leverage the Notion of Sustainable Diet Within Socio-Technical and Socio-Ecological Systems | Stakeholders as Coalition of Actors | | | | | |
|---|---|---|---|---|---|---|---|---|
| | | | Consumers | Environmental Action Partnerships | Producers/ Farmers | Socio-Cultural NGOs | Value Chain Financing Specialist | Youth/Women Groups |
| Niche Micro-level: Stimulation of local experiments refers to An inclusive, practice-based, and challenge-led socio-technical initiative designed to promote system innovation through social learning under conditions of uncertainty and ambiguity | Policies supporting niches | Elaborating effective schemes for embarking in: NUS crop production value chain Promotion of crop and dietary diversity | ✓ | | ✓ | | ✓ | ✓ |
| | Support for the creation of niche networks between various stakeholders | Establishing communication channels between stakeholders: Fostering access to credit/value chain establishment at small-scale levels | ✓ | ✓ | ✓ | | ✓ | |
| | | Mainstreaming awareness of biodiversity loss and the cascading impacts across socio-ecological systems | ✓ | ✓ | ✓ | | | |
| | Monitoring food choice determinants | Understanding the shift from traditional to modernity through lived experience Social media analysis of food choice | ✓ | | ✓ | ✓ | | ✓ |
| | | Public co-funding of bottom-up initiatives: small-scale traditional (Gogo, meaning Grandmother) food canteens | ✓ | ✓ | ✓ | ✓ | ✓ | |
| | Normalizing environmental impacts of land use shifts | Systematic mapping of sugarcane and forestry land use | | | | ✓ | | |
| | Supervision of sustainable beneficiation of natural capital | Improve cross-sectoral evidence on natural capital sustainability | | ✓ | ✓ | | | ✓ |
| Regime Mesolevel: The Food Environment that needs to be changed, but consists of dominant actors, institutions, practices, and presumed shared objectives | Support for the expansion of a targeted sector | Rural agri-tourism | ✓ | ✓ | ✓ | ✓ | ✓ | |
| | | Education | ✓ | ✓ | ✓ | ✓ | | ✓ |
| | Policies limiting the power of regimes | Transparency of lobbying processes | ✓ | ✓ | ✓ | ✓ | | ✓ |
| | | IKS inclusion | | | | | | |
| | Promotion of technical or resource diversity | Public R&D investments and subsidizing private R&D in agroecological intensification | | ✓ | ✓ | | | |
| | Regulating unhealthy consumption activities | Taxes or tradable permits, command-and-control of products such as sugar tax, fast food | ✓ | | ✓ | | | |
| Landscape Meta level Economic, ecological, socio-political, conditions, e.g., the South African Constitution that provides the context to drive niche experiments and actions | Promotion of civic debate | Public participation in policy development (round tables). | ✓ | ✓ | ✓ | ✓ | | ✓ |
| | Information provision | Informative campaigns for consumer behavior | ✓ | | | | | ✓ |
| | Creation of informed debate | Supporting public participation in setting the policy agenda | ✓ | ✓ | ✓ | ✓ | | ✓ |
| | Developing policy integration (technology, environment, consumers) | Making one ministry responsible for coordinating all initiatives and policies concerning long term sustainability transition | ✓ | ✓ | ✓ | | | |

[3] NUS: Neglected and Underutilized Species. IKS: Indigenous Knowledge Systems.

## 4. Discussion

### 4.1. Understanding the Mechanism Used to Co-Design Change Towards Sustainable Diets

The study uses an interactive facilitation process among stakeholders to envision and co-design a future state of the food system by prioritizing the research focus for the SHEFS consortium that ought to be both sustainable and healthy for the smallholder system and previously disadvantaged group in South Africa. To this end, policymaking would require interdisciplinary evidence capable of leveraging the outcomes of future implementation efforts. Sustainable food consumption occurs in the nexus between the national context

and private individual lifestyle [67]. Similarly, the synergy required between the perception of health and sustainability differs across contexts [74]. Therefore, taking these multiple domain relationships into account, we have shown that, due to the inherently complex nature of socio-technical and socio-ecological systems within which sustainable diets must be embedded, most intervention strategies are likely to take effect by way of multiple mechanisms, although it remains an empirical and/or contextual issue whether one mechanism is primary, and others are ancillary. In effect, it is also likely that the same mechanism might be involved in the operation of multiple implementation strategies as shown in the causal loop diagrams (CLDs). To gain clarity on the emergent outcomes of the CLDs [67], these were unpacked in a logical framework comprising the following elements: Intervention → Causal pathway → Expected outcomes → Relevance to global goals. The logical analysis acknowledges that these transformative processes ought to occur through multi-dimensional mechanisms—comprising institutional rules, economic requirements, multi-level political negotiations as well as social and cultural rules and expectations—from the local to the global scale. Herein, following Lewis et al. [90], we consider "mechanisms" as the processes or events through which an implementation strategy functions to achieve desired outcomes. Careful considerations were taken to ensure that each strategic intervention is well-specified and judiciously linked to its corresponding mechanisms in a coherent manner. This is because underspecified strategies can potentially leave the interdisciplinary research space vulnerable to inappropriately synthesizing data across studies [91,92]. Herrfahrdt-Pähle et al. [93] used the example of successful water governance in post-apartheid South Africa to emphasize that different phases of transformation require different features and capacities. It is to ensure such coherence that the interventions proposed by the stakeholders in the present study were derived through causality and system loops, and thereafter embedded in the niche–regime–landscape transition framework.

### 4.2. Emergent Entry-Points for Transformative Evidence Building

Five inter-linked areas have emerged from the stakeholder engagement process, which can be used to define priority entry points to build evidence-based policies that align with sustainable and healthy food systems. The first one refers to breaking away from the legacy of apartheid by advocating transformative governance that acknowledges the pervasive disconnect between, on the one hand, the microlevel socio-political reality of the previously disadvantaged, parochial evidence synthesis and practice and, on the other hand, the positive expectations of the macro-level landscape—Bill of Rights in the South African Constitution [94,95]—but which is crippled with counterintuitive effects due to emphasis on the sectoral development agenda that results in decades-long pervasive delays to alleviate the smallholder sector. Successful political transformation, that is the shift to democratic South Africa, has not realistically ensured a new normal in terms of social and economic transformation, especially for historically underprivileged smallholder food producers, as the country remains the most unequal society [96]. Second, there was consensus that the critical challenges to be acknowledged in realizing intervention efforts requires multi-dimensional evidence-based policy solutions, similar to a Context–Mechanism–Outcome configuration [30], that exhibit dynamics of three functional properties of a knowledge economy about wider transformative processes: Identifying positive feedback patterns through education to accumulate multi-functional capabilities, nurturing evidence synthesis for improved practice by way of informational and adaptive policymaking, and empowerment of youth through grass-root actions to capacitate social cohesion, dignity, and identity construction. Third, the development and governance of the smallholder food sector ought to foster environmentally sustainable and resilient food systems that can mitigate the impact of unintended consequences of policies that promote commercialization/intensification of food production to the detriment of subsistence farming, household food security, food crop diversity, and dietary diversity. Although huge transformative efforts have been achieved to break the "Success to the successful" apartheid system archetype, the transitioning achieved in the last decades is crippled by the "Shifting the burden" archetype due

to systemic delays across sectors. Socio-political shocks can be windows of opportunity, but the process needs to be navigated. Fourth, the ensuing dietary diversity could be partly aligned with the needs of providing healthy diets. In addition, to further essential nutrition actions, supportive educational measures promoted by health literacy ought to guide social ambitions towards food choice for well-being by promoting sustainable and healthy behavioral shifts when aspiring to transition from a traditional to a modern lifestyle. Fifth, proper recognition of the importance of environmental literacy should be actioned by mainstreaming awareness of biodiversity loss and its negatively reinforcing impacts across socio-ecological systems. At the same time, environmental literacy could improve cross-sectoral evidence on natural capital sustainability, and support the expansion of entirely novel sectors such as agri-tourism at the smallholder level.

Key features need to be mobilized for transformation for a sustainable diet innovative system. At the cognitive level, this would require change at the individual level as well as in broader social units and communities of practice, e.g., through transdisciplinary participation and collaborative governance. Structurally, the combination of different types of knowledge, and preserving and making knowledge available, ought to evolve as conditions for identifying sound policy instruments to improve the ability to deal with transformative change. Swensson et al. [97] reviewed the role of the regulatory framework in the facilitation of public food procurement for the implementation of socio-economic objectives through public procurement. Although such policy instruments have been adopted in various countries within this specific context, comprehensive analysis is still lacking in the food policy context. In this paper, we have a number of measures in the meso-level policy regime.

The process used for interactive facilitation, and subsequently embedding the emergent outcomes in the MLP of the transition systems framework, has contributed to a broader reflection on deliberatively strategizing, shaping, and modulating sustainable diet pathways towards desirable individual and societal outcomes, in full awareness of the scale, influence, and urgency of the effort required. The co-designing process used to problematize sustainable diet transition, as reported in the current work, set the pace in developing actionable research for the project and its Theory of Change.

## 5. Conclusions

The emergent outcomes of the current work demonstrate the complex nature of sustainable diet transitioning by highlighting the multiple interdependencies across sectors and cross-scale dynamics. Intervention strategies to inform policies, therefore, cannot be designed as stand-alone approaches. Rather, emphasis should be placed on co-evolutionary sets of measures to inform decision-making for the real world. This work examines key issues raised by stakeholders' considerations by combining causal mechanisms leading to sustainable diets and embedding the proposed strategies in a multi-level perspective of the transition theory. The mapping of these issues builds knowledge from, and for, practice, by linking different perspectives, including dietary diversity, sustainable beneficiation of natural capital, and food choice for well-being, via the "Intervention → Causal pathway → Expected outcomes → Relevance to global goals" mechanism. A similar approach could be applied in other contexts to problematize sustainable diet transitioning. We have set out five major emergent outcomes of the co-designing process with stakeholders. Despite the very wide knowledge base, disciplines, and methodological differences involved in framing sustainable diets in South Africa, we show how the different levels of the sustainable diet innovation systems (landscape, regime, and niche) could interact to pave the way for initiating such transformations and which key features (cognitive, structural, and agency-related) are mobilized for transformation.

**Author Contributions:** Conceptualization: N.S. and R.S.; methodology: N.S. and R.S., formal analysis: N.S. and R.S.; investigation: N.S., C.B.C., and R.S.; writing—original draft: N.S. and R.S.; writing—review and editing: N.S., R.D., C.B.C., T.M., A.D.D., A.T.M. and R.S.; visualization: N.S.,

P.S., and R.S.; project administration: R.D. All authors have read and agreed to the published version of the manuscript.

**Funding:** This research is part of the SHEFS—an interdisciplinary research partnership forming part of the Wellcome Trust's funded Our Planet, Our Health program, with the overall objective to provide novel evidence to define future food systems policies to deliver nutritious and healthy foods in an environmentally sustainable and socially equitable manner. This research was funded by the Wellcome Trust through the Sustainable and Healthy Food Systems (SHEFS) Project (grant no. 205200/Z/16/Z).

**Informed Consent Statement:** Written informed consent was obtained from all subjects in the study as per the ethical clearance guidelines of the University of KwaZulu Natal, South Africa. Informed consent was obtained from all subjects involved in the study.

**Conflicts of Interest:** The authors declare no conflict of interest.

## Appendix A

Action learning: Conceptualization of the interrelationships of the sustainable and healthy food systems in South Africa

Revans's [98] action learning concept was applied to capture the learning outcome of the workshop. The concept specifies that unless problems are open to a purely technical solution, there is more learning to be grasped before action is taken by those involved with an issue. It constitutes (i) System alpha, which centers on the investigation of the problem, examining the external context, structural values, and available resources; (ii) System beta focuses on problem resolution, through decision cycles of negotiation and reflection and, (iii) System gamma concerns the participant's cognitive framework, their assumptions, and prior understanding, and is concerned with learning as experienced by each stakeholder type. The three systems, alpha, beta, and gamma are not linear or sequential, nor are they entirely discrete. All types of stakeholders possess "Programmed Knowledge", which can only help individuals or organizations up to a point. However, dealing with change requires greater insight and this is gained by posing "Questions". Therefore, "Learning" then becomes a function of acquiring programmed knowledge and combining it with questioning insight, expressed by Reg Revan's Learning Equation: L (Learning) = P (Programmed Knowledge) + Q (Questioning Insight)

The principal interest in developing effective learning to achieve adaptation and deal with change was to focus on Q, Questioning Insight. It is the ability to exploit the questioning insight that would give rise to the interrelated multiple perspectives in co-designing the SHEFS program objectives. Action learning recognizes that, in the absence of insight, the use to which an abundance of programmed knowledge may be put is limited. Problems and opportunities are treated by leaders (in funded research, these refer to program managers/principal investigators) who must be aware of their value systems, differing between individuals (i.e., stakeholders), and the influences of their past personal experiences [98].

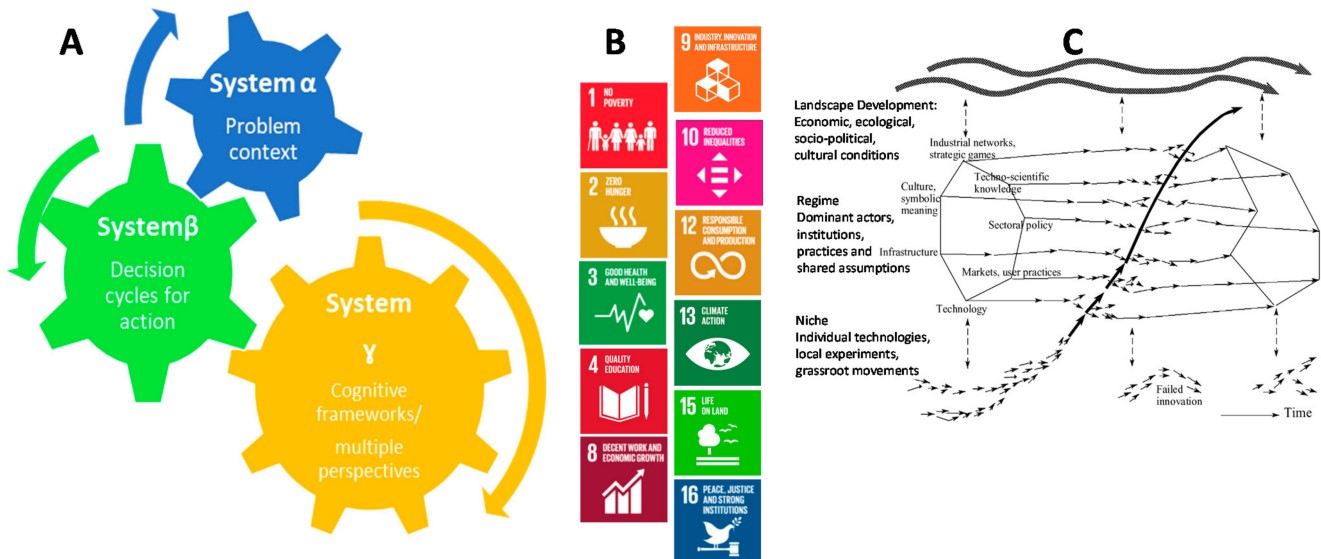

**Figure A1.** Describes the workshop process and causal loop diagram. (**A**) Systems alpha: Context-specificity consideration. Systems beta: Which components and/or lenses to consider to optimally intervene and focus on the investigation of the problem. System gamma: Focus on the learning, i.e., how to intervene collectively based on the dimensions identified. The three systems are best understood as a whole, with interlocking yet overlapping parts [99]. (**B**) Unpacking the systemic interactions of the problem context through a logical framework that identifies relevance with the SDGs. (**C**) Developing leverage points, identified from processes in A, to generate evidence capable of stimulating the policymaking process through alignment within a multiple-level perspective (niche–regime–landscape) of the transition theory. Part C of the diagram is adapted from Geels [51].

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
