# Peer review of "Visioning a Food System for an Equitable Transition towards Sustainable Diets—A South African Perspective"

_sustainability, doi:10.3390/su14063280_

Round 1

Reviewer 1 Report

 The authors could cite some relevant papers from journals such as Global Food Security, Food Quality and Preference etc. The references should be up-to-date.  It is not clear why the authors have simultaneously applied the bibliometric and systematic review methods. The methodology (section 2) needs to be more specific and concise.  The conclusion is loosely written. It must be re-written in light of the main findings and purpose.  The authors should justify the need and importance of the work in the introductory section.  What is the main contribution of the paper? Is it a review type or original work?

Author Response

Reviewer 1

  1. The authors could cite some relevant papers from journals such as Global Food Security, Food Quality and Preference etc. The references should be up-to-date.

The most relevant and recent papers from Global Food Security and Food Quality and Preference have been cited.

  1. It is not clear why the authors have simultaneously applied the bibliometric and systematic review methods. The methodology (section 2) needs to be more specific and concise.

 This paper uses an exploratory approach to show the co-design process applied at the beginning of the SHEFS project. The emergent issues raised during the interactive facilitation, consisting of the stakeholder meeting and the de-briefing analysis thereafter, were used to guide further steps (Concept Note development) in the co-production of interdisciplinary evidence to inform sustainable diet policies. To maximize the robustness of this paper’s outcome, insights from scientific realism are used. Thus, on the one hand, the bibliometric method is applied to corroborate with the systemic relationships identified in the causal loop analysis. On the other hand, the systematic review is used to support the argument for creating a sustainable diet innovation system (SDIS) based on the premise that sustainable diet transitioning (SDT) can be viewed from the socio-technical perspective. The papers reviewed in the scoping review aid in building the framework in the current paper. Viewed altogether, the causal loop analysis, the bibliometric analysis, the scoping review, and the multiple level perspective of the transition systems theory serve as a means of triangulation to conceptualize the emerging and co-evolving issues that need to be considered to inform policies on SDT.

The conclusion is loosely written. It must be re-written in light of the main findings and purpose.

The Conclusion section has been re-written by the other changes carried out throughout the paper.

  1. The authors should justify the need and importance of the work in the introductory section.

Based on the reviewers’ comments, the notion of the sustainable diet innovation system (SDIS) has now been used in the introduction to justify the need for the interdisciplinary effort required and the importance of the current work.

  1. What is the main contribution of the paper? Is it a review type or original work?

The paper essentially uses qualitative systems dynamics, which is an aspect of systems theory, as a method to frame the dynamic behavior of sustainable diet transitioning. This step is used to emphasize the realist perspective that must be considered when generating evidence to inform policies that deal with complex systems, herein food, environment, health, consumer aspirations, socio-political history. The positive and negative feedback mechanisms explain the nature of the issues that need to be accounted for when intending to devise evidence-based policies.

System archetypes were applied to show the impact of the socio-political history and the systemic delays that it has caused in achieving the desired transformative changes across sectors in South Africa.

Therefore, the implementation of the research output needs to be developed in such a way that opportunities for capabilities are harnessed across several actor domains; indicating that we have to acknowledge the need to shift from the realm of Mode 1 research and extend the epistemic domain to Mode 2 research by making it inclusive towards practitioners, beneficiaries and various other stakeholders as the need of the project and policymaking arises. The present work represents a conceptual framework, from a realist perspective, demonstrating how this shift can be achieved; hence justifying the transitions standpoint. And, the triangulation method was used to support the justifications. As a means to connect the local to the global context, we use the United Nations Sustainable Goals to show how intervention pathways are expected to deliver expected outcomes in line with the relevant targets. In this regard, we view the paper as an original work.

Reviewer 2 Report

Dear Authors,

The subject of your article (access to dietary diversity, sustainable beneficiation of natural capital, and food choice for well-being) is so actual and, unfortunately, will remain actual many years from now.

The research released in the article could be useful for decision government people to vision a better law release to help people in need.

The article has an interesting view of data, is well documented, and well-illustrated for a better understanding of the presented research.

The paper is well described and the methods used are scientifically appropriate. Only a few points should be addressed by the authors:

  • ” Visioning a food system for the equitable transition towards sustainable diets” – in my opinion, the title of the paper should be reformulated to highlight the subject of the research and to specify the location of the proposed system – in this form, is too general
  • Please, pay attention to the affiliation of the author – please, follow the template
  • For such elaborated research, the size sample of ”39 stakeholders” could be considered too small, but due to the competence and the diversity of the people choose for the ”sample”, it should be considered as appropriate
  • The research is well documented and argumented – the authors analyzed very well all the aspects of the emerging problems regarding food insecurity
  • There are a few and small spelling issues all over the paper which could be resolved using the grammarly.com
  • The paragraph between lines 409 and 412 should be written using the same font as the rest of the paper
  • Please, pay attention to the references section: the 46 and 47 are the same! Please change the numbering of the references also, in the paper, from that point!
  • Please, indicate the year for the references 3 and 32, and add more info about reference numbers 13, 83, 84, 95, and for 56 and 66 (the date of access)
  • Please, verify the reference list – all references should be written according to the template

Thank you!

Author Response

Manuscript ID: sustainability-1594778
Type of manuscript: Article

Reviewer 2

1            “Visioning a food system for equitable transition towards sustainable diets” – in my opinion, in the title of the paper should be reformulated to highlight the subject of the research and to specify the location of the proposed system – in this form, is too general

The title of the paper has been modified to specify the South African context and now reads as follows: “Visioning a food system for an equitable transition towards sustainable diets – a South African perspective’.

2            Please, pay attention to the affiliation of the author – please, follow the template

Author affiliation has been revised and corrected. The use of the template for the author section has been updated.

3            For such an elaborated research, the size sample of ”39 stakeholders” could be considered too small, but due to the competence and the diversity of the people choose for the” sample”, should be considered as appropriate 

Indeed, the targeted sample size should have been 50 for the interactive facilitation that uses a systems approach. However, some of the participants could not attend the workshop. The shortcomings, if any, were counteracted during the peer debriefing session on the following day.

4            The research is well documented and argumented – the authors analyzed very well all the aspects of the emerging problems regarding food insecurity

Special focus was placed on the analysis of the causal loop diagrams, derived from the systems theory, to identify emerging interrelationships and relevant domains characterizing the sustainable diet system.

5            There are a few and small spelling issues all over the paper which could be resolved using the www.grammarly.com

The spelling issues have been checked by using Grammarly.com.

6            The paragraph between lines 409 and 412 should be written using the same font as the rest of the paper 

The font for Table footnote has been applied to the lines corresponding to 409 to 412.

7            Please, pay attention to the references section: the 46 and 47 are the same! Please change the numbering of the references also, in the paper, from that point!

The double-entry of the same reference has been rectified and the references numbering updated.

8            Please, indicate the year for the references 3 and 32, and add more info about the reference numbers 13, 83, 84, 95, and 56 and 66 (the date of access)  

The reference list has been updated to reflect the required changes.

Round 2

Reviewer 1 Report

Accept in present form.